# Pre-stimulus phase and amplitude regulation of phase-locked responses are maximized in the critical state

**Arthur-Ervin Avramiea[1†], Richard Hardstone[1,2†], Jan-Matthis Lueckmann[1,3], Jan Bím[1,4], Huibert D Mansvelder[1], Klaus Linkenkaer-Hansen[1*]**

[1]Department of Integrative Neurophysiology, Center for Neurogenomics and Cognitive Research (CNCR), Amsterdam Neuroscience, Amsterdam, Netherlands; [2]Neuroscience Institute, New York University School of Medicine, New York, United States; [3]Technical University of Munich, Munich, Germany; [4]Czech Technical University in Prague, Prague, Czech Republic

**Abstract** Understanding why identical stimuli give differing neuronal responses and percepts is a central challenge in research on attention and consciousness. Ongoing oscillations reflect functional states that bias processing of incoming signals through amplitude and phase. It is not known, however, whether the effect of phase or amplitude on stimulus processing depends on the long-term global dynamics of the networks generating the oscillations. Here, we show, using a computational model, that the ability of networks to regulate stimulus response based on pre-stimulus activity requires near-critical dynamics—a dynamical state that emerges from networks with balanced excitation and inhibition, and that is characterized by scale-free fluctuations. We also find that networks exhibiting critical oscillations produce differing responses to the largest range of stimulus intensities. Thus, the brain may bring its dynamics close to the critical state whenever such network versatility is required.

**\*For correspondence:**
klaus.linkenkaer@cncr.vu.nl

†These authors contributed equally to this work

**Competing interests:** The authors declare that no competing interests exist.

## Introduction

Understanding how neurons coordinate their activity to produce emergent dynamics and behaviors is crucial to understanding how the brain gives rise to conscious perception. An influential approach for elucidating neural correlates of consciousness (*Dehaene and Changeux, 2011*; *Engel and Singer, 2001*; *Tononi and Koch, 2008*) has been that of threshold-stimulus detection (*Li et al., 2014*; *Linkenkaer-Hansen et al., 2004*; *Wyart and Tallon-Baudry, 2009*). In these tasks, subjects are given a stimulus that is set at the edge of perception, meaning that on 50% of trials subjects perceive it, and on 50% of trials they do not. In trials where the stimulus is perceived, primary sensory regions exhibit larger event-related potentials (*Pins and Ffytche, 2003*), larger event-related desynchronization (*Vidal et al., 2015*), and stronger phase-locking to the stimulus (*Palva et al., 2005*). However, what causes some trials to evoke stronger ERPs, event-related desynchronization, or phase-locking than other trials—with dramatic consequences for perception—cannot be explained by differences in the stimuli, which are held identical throughout the threshold-stimulus detection experiments. The alternative is that awareness of stimuli depends on brain processes that unravel around the time of the stimulation, for example that pre-stimulus neuronal activity biases stimulus processing and conscious perception (*Aru et al., 2012*).

One candidate mechanism for gating stimulus processing is alpha oscillations (*Jensen and Mazaheri, 2010*), which inhibit unwanted stimuli from reaching consciousness (*Capilla et al., 2014*; *Händel et al., 2011*; *Jensen et al., 2012*; *Jensen and Mazaheri, 2010*; *Thut et al., 2006*). Fluctuations in alpha oscillations, possibly modulated by attention (*Macdonald et al., 2011*), may determine

whether a stimulus is perceived or not. This was confirmed in threshold-stimulus experiments relating higher pre-stimulus alpha activity to decreased stimulus detection (*Ergenoglu et al., 2004*; *Hanslmayr et al., 2005*; *Linkenkaer-Hansen et al., 2004*). In addition to amplitude, the pre-stimulus alpha phase also predicts stimulus awareness (*Busch et al., 2009*; *Mathewson et al., 2009*; *VanRullen, 2016*).

Regardless of the attentional state, neuronal oscillations in the alpha band exhibit spontaneous amplitude fluctuations. However, these fluctuations are not random, but exhibit a specific temporal structure: autocorrelations with scale-free decay that extends up to time scales of hundreds of seconds (*Linkenkaer-Hansen et al., 2001*), also called long-range temporal correlations (LRTC). This scale-free character of the oscillations emerges in neuronal networks with balanced excitatory and inhibitory forces (*Poil et al., 2012*) and is considered a sign of critical-state dynamics (*Bak et al., 1987*; *Chialvo, 2010*). Thus, oscillations exhibiting LRTC are referred to as 'critical oscillations'. Criticality in the brain has also been associated with scale-free neuronal avalanches, and different computational models have used this link to indicate that critical dynamics are beneficial for processing information (*Beggs and Plenz, 2003*), for example evoked responses show the largest dynamic range (*Gautam et al., 2015*; *Kinouchi and Copelli, 2006*; *Larremore et al., 2011*) and this has subsequently been confirmed experimentally (*Gautam et al., 2015*; *Shew et al., 2009*). However, it is not known whether critical-state dynamics of alpha oscillations is beneficial for stimulus processing. Importantly, it remains entirely unknown whether pre-stimulus bias of post-stimulus processing depends on brain dynamics being close to the critical point.

To answer this question, we consider the network response in terms of phase-locking to the stimulus, which is thought to play a role in reorganizing neuronal networks for efficient stimulus processing (*Hanslmayr et al., 2005*; *Min et al., 2007*; *Sauseng et al., 2007*; *Voloh and Womelsdorf, 2016*). We rely on an extension of a model of critical oscillations (CROS) (*Poil et al., 2012*), which exhibits dynamics comparable to human M/EEG (*Dalla Porta and Copelli, 2019*). We use the model to investigate the network phase-locking response as a function of network connectivity, stimulus size, and pre-stimulus alpha amplitude and phase. We then related short-term fluctuations in information processing in these networks, with a long-term characteristic of the oscillatory dynamics: their level of criticality. To evaluate the criticality of alpha oscillations, we used detrended fluctuation analysis (DFA, see Materials and methods), which can discriminate between random fluctuations in the amplitude of oscillations and critical oscillations with long-range temporal correlations. In addition, DFA has the advantage, over other measures of criticality such as neuronal avalanches, that it can be estimated noninvasively at electrode level. This will allow future experiments with EEG/MEG recordings to test whether for example the criticality of sensory cortices biases stimulus processing. We show that pre-stimulus amplitude and phase regulation occurs for critical, but not for subcritical or supercritical networks. Thus, critical-state dynamics in a neuronal network is associated with versatile functions, allowing it to flow between low- and high-responsivity states on sub-second time scales. In addition, we show that a critical network can differentiate the widest range of stimulus sizes. Together, our results suggest that understanding how close neuronal networks are to criticality is essential for understanding their function.

## Results

### Unstimulated network produces multi-level criticality

To test the effect of critical oscillations on neuronal network functionality, we adapted a model of ongoing neuronal activity (CROS) (*Poil et al., 2012*) (see Materials and methods). CROS consists of 75% excitatory and 25% inhibitory integrate-and-fire neurons arranged in a $50 \times 50$ grid (*Figure 1—figure supplement 1A*). The two parameters that need to be set when creating a network are the percentage of neurons within a local range (circular radius four neurons) that each excitatory and each inhibitory neuron connects to (*Figure 1—figure supplement 1A*). The firing rate grew monotonously for both excitatory and inhibitory neurons, as we increased excitatory connectivity (*Figure 1—figure supplement 1B,C*). Networks oscillated in the 8–16 Hz range (*Figure 1—figure supplement 1D*), with the power of the oscillation increasing as we move towards an excitation-dominated regime (*Figure 1—figure supplement 1E*). Importantly, the model exhibits multi-level criticality, with spatial and temporal power-law scaling behavior in network activity, which arises through a

common mechanism of balanced E/I connectivity (*Figure 1—figure supplement 1F–J*). To illustrate the importance of the level of criticality on network function, we will show detailed data from three networks throughout the paper: a subcritical (45% excitatory/90% inhibitory connectivity - *blue*), a critical (60%/75% - *green*), and a supercritical one (75%/60% - *red*)(e.g., *Figure 1E*).

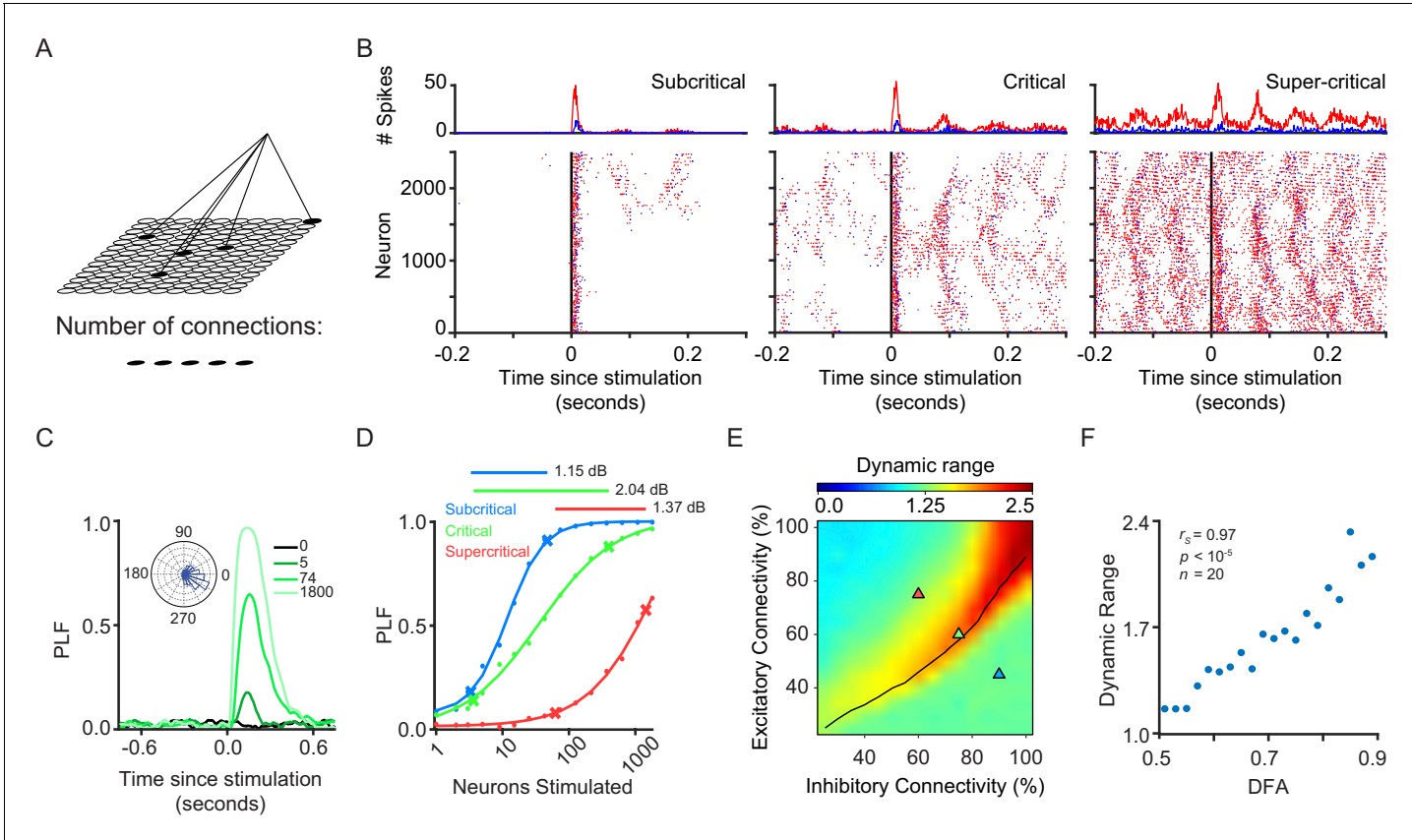

**Figure 1.** Dynamic range is maximized when network exhibits critical-state dynamics. (**A**) A stimulus is connected to a random subset of excitatory neurons on the grid. (**B**) We stimulated 1800 excitatory neurons in three example subcritical, critical and supercritical networks. The top plots show the total number of excitatory (*red*) and inhibitory (*blue*) spikes with millisecond resolution. Plots below show individual spikes for all 2500 excitatory and inhibitory neurons in the networks. In all cases, the response to the stimulus is oscillatory with a periodicity characteristic of an alpha (8–16 Hz) rhythm. (**C**) The phase-locking factor at a time point post-stimulus is calculated as the normalized vector sum of phase angles across trials (inset left). A value of 1 corresponds to all trials having the same phase, a value of 0 corresponds to a uniform distribution of phases. Post-stimulus phase-locking response to a stimulus is dependent on the size of the stimulus. Shown for different stimulus sizes in a critical network. (**D**) Dynamic range was calculated as the orders of magnitude of neurons stimulated between the 10th and 90th percentile, of a sigmoid fit to the phase-locking response, at a time point where the network shows response to the stimulus (150 ms post-stimulus). Shown for example sub-critical (*blue*), critical (*green*) and super-critical (*red*) networks. (**E**) Dynamic range is dependent on E/I connectivity balance. The *black line* indicates critical neuronal avalanches, as measured when the networks were not stimulated. The networks used to illustrate the functionality of different criticality regimes are shown as triangles on the phase-space with the colors of sub-critical (*blue*), critical (*green*) and super-critical (*red*). (**F**) Dynamic range of phase-locking response increases with long-range temporal correlations (LRTC) at 150 ms post-stimulus (Spearman correlation). Networks binned based on average DFA for each of the 256 excitatory/ inhibitory connectivity parameter combinations, and dynamic range calculated as the mean of networks in that bin.

The online version of this article includes the following figure supplement(s) for figure 1:

**Figure supplement 1.** The CROS model produces critical state dynamics in the form of neuronal avalanches and long-range temporal correlations (LRTC) of oscillations, when excitation is balanced by inhibition.

**Figure supplement 2.** Critical networks exhibit maximum dynamic range of phase-locking factor at multiple post-stimulus latencies.

**Figure supplement 3.** Critical networks show maximal dynamic range of phase-locking factor.

## Dynamic range of phase-locking response is strongest for critical networks

Previously, it was shown that criticality maximizes the dynamic range of evoked firing-rate responses (*Gautam et al., 2015*; *Kinouchi and Copelli, 2006*; *Larremore et al., 2011*). However, event-related responses in large oscillatory networks also reflect a reset in the oscillatory phase following stimulation, leading to the phase-locking of the oscillation to the stimulus (*Makeig et al., 2002*; *Min et al., 2007*; *Palva et al., 2005*). As such, the firing rate and phase-locking responses reflect fundamentally different aspects of network activity, for example evoked firing-rate responses may increase the overall network activity without affecting the phase of oscillatory activity in a particular frequency band. In fact, evoked firing-rate responses are not an appropriate measure of network response in networks with strong oscillations: they fluctuate continuously with the phase of the ongoing oscillation, making it difficult to judge the firing-rate response in absolute values, that is without relating it to the ongoing oscillation. In that respect, the phase-locking response is a more appropriate way to detect changes in network activity occurring due to external stimulation.

To test whether criticality maximizes the dynamic range also for phase-locked oscillatory responses, we attached a stimulus to $n$ excitatory neurons distributed randomly on the grid (*Figure 1A*). The stimulus had the same weight as any of the excitatory to excitatory synapses in the network ($w_{input} = w_{EE} = 0.0085$), and lasted for one time step (1 ms). We found that the stimulus response in our networks is indeed oscillatory, which may result in stimulus phase-locking (*Figure 1B*). We then investigated how changing the stimulus size, $n$, altered the phase-locking response of a network to the stimulus. In *Figure 1C*, we can see that a critical network increases the level of post-stimulus phase-locking as stimulus size is increased. This shows that the CROS model is capable of showing oscillatory responses to a stimulus, as has been seen in human subjects (*Palva et al., 2005*). We calculated the dynamic range, which gives an indication of how well a network can give differing responses to stimuli of different sizes (see Materials and methods). Critical networks show a wider dynamic range than their sub-/super-critical counterparts (*Figure 1D–F*), and this effect is robust over a wide range of post-stimulus latencies (*Figure 1—figure supplement 2*). Interestingly, in the case of supercritical networks, the strong, ongoing oscillatory activity prevents the phase-locking response from saturating (PLF ~1), even at the largest stimulus sizes (*Figure 1D*, *Figure 2—figure supplement 1*). Another factor that can shape stimulus response, other than stimulus size, is the strength of the external stimulus. We found that an 8-fold increase in the external input strength was sufficient to bring the PLF of supercritical networks to saturation, for the largest stimulus size (*Figure 1—figure supplement 3B*). However, such a large increase in the input strength also leads to a stronger PLF for the smaller stimulus sizes in the case of subcritical and critical networks, which means they no longer show weak PLF responses reducing their dynamic range. Of all tested strengths and networks, the maximal dynamic range occurs for critical networks at a multiplier ~1 (*Figure 1—figure supplement 3C*). Overall, our results show that critical networks discriminate the widest range of stimulus sizes through their post-stimulus phase-locking response.

## Pre-stimulation amplitude influence on phase-locking requires critical dynamics

The ongoing oscillations showed a phase-locking response in the post-stimulus period (~65–250 ms) to the stimulus, which depended on the criticality of the network (*Figure 2A,B*), as well as on the stimulus size (*Figure 2—figure supplement 1*). Considering the complex variation of activity in a critical network, we wondered whether critical networks function differently depending on their instantaneous state. For this purpose, we assessed the network phase-locking response to a stimulus of 5 neurons, and related it to the pre-stimulus amplitude averaged over the −150 to −50 ms time range, in the 8–16 Hz frequency band. We found that critical networks exhibit strong negative pre-stimulus amplitude regulation (*Figure 2C–E*), whereas sub-critical or super-critical networks show no pre-stimulus amplitude regulation (*Figure 2D,E*). We confirmed this result statistically by showing that pre-stimulus amplitude regulation was more prominent for networks with stronger LRTC (*Figure 2F*, *Figure 2—figure supplement 2*).

To test whether this function is generic to criticality and not specific to the chosen stimulus size, we investigated the three combinations of excitatory/inhibitory connectivity and found that the critical network—unlike the sub-/super-critical networks—showed significant pre-stimulus amplitude

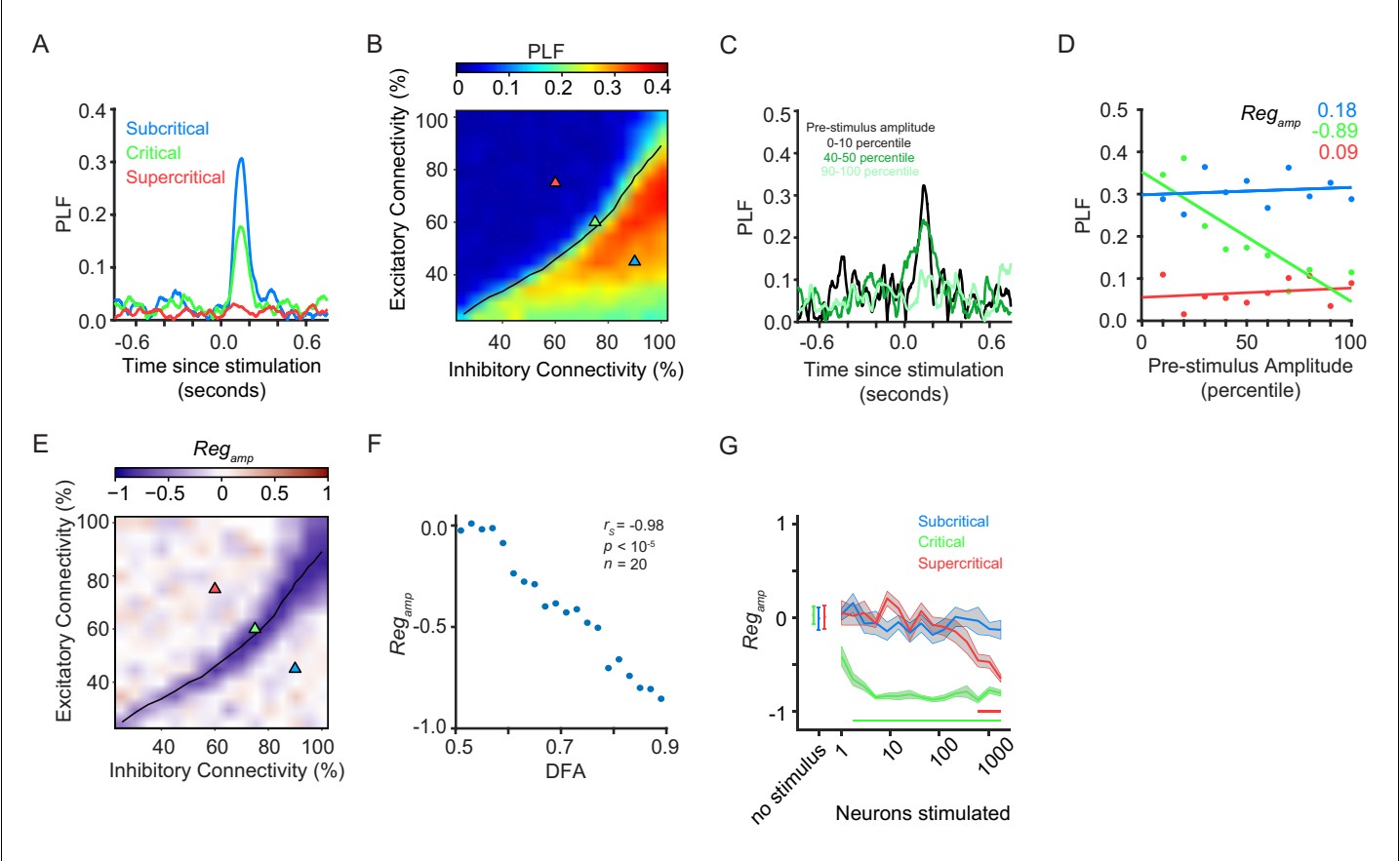

**Figure 2.** Pre-stimulus amplitude regulation of response requires critical-state dynamics. (**A**) Ongoing oscillations phase-lock to stimulus. Example shown for five neurons stimulated, for sub-critical (*blue*), critical (*green*), and super-critical networks (*red*). (**B**) Phase-locking response at 150 ms post-stimulus depends on the balance between excitation and inhibition, being highest for sub-critical networks, lowest for super-critical networks, and with intermediate values in the critical regime. The locations of the three example networks are highlighted on the phase space by triangles with corresponding colors. *black line* indicates critical neuronal avalanches. (**C**) To calculate the PLF response across trials with similar pre-stimulus amplitude, we split trials in 10 bins based on the pre-stimulus amplitude in the −150 to −50 ms time range, for the 8–16 Hz frequency band. Trials belonging to different pre-stimulus amplitude bins show different phase-locking response post-stimulus. Color indicates the percentile interval of the pre-stimulus amplitude bin. (**D**) Power of pre-stimulus oscillation can alter the phase-locking response of a network to the stimulus, in the presence of critical oscillations. Shown for example sub-critical (*blue*), critical (*green*), and super-critical (*red*) networks. Pre-stimulus amplitude regulation, $Reg_{amp}$, was computed as the Spearman correlation coefficient between the pre-stimulus amplitude percentile bins, and the PLF of the trials in each bin. (**E**) The strength of pre-stimulus amplitude regulation is dependent on a balance between excitation and inhibition in terms of connectivity. The *black line* indicates critical neuronal avalanches, as measured when the networks were not stimulated. Here shown for a stimulus size of 5 neurons. (**F**) Pre-stimulus amplitude regulation ($Reg_{amp}$) is more prominent for stronger LRTC (DFA) at 150 ms post-stimulus (Spearman correlation). DFA was measured for the non-stimulated networks, and $Reg_{amp}$ was measured for the same networks, but with a five neuron stimulus. Networks binned based on DFA as in *Figure 1F*. (**G**) Critical networks exhibit pre-stimulus amplitude regulation for the widest range of stimuli. Supercritical networks also show regulation, but only at very high stimulus sizes. mean ± sem corresponding to 10 individual networks with the same excitatory/inhibitory connectivity as the selected sub-critical (*blue*), critical (*green*), super-critical (*red*) networks. Horizontal lines at the bottom of the plot show the range of stimuli for which the networks show significant pre-stimulus regulation (two-sided t-test, null hypothesis mu = 0, Bonferroni corrected for 3 × 16 comparisons).

The online version of this article includes the following figure supplement(s) for figure 2:

**Figure supplement 1.** The phase-locking factor depends on excitatory-inhibitory connectivity, and increases with the stimulus size.

**Figure supplement 2.** Critical networks exhibit maximum amplitude regulation at multiple post-stimulus latencies.

**Figure supplement 3.** Critical networks exhibit robust pre-stimulus amplitude regulation for a wide range of input stimulus strengths.

modulation for a wide range of stimulus intensities (*Figure 2G*). Changing the weight of the external input also affects the range of stimuli over which networks show pre-stimulus amplitude regulation (*Figure 2—figure supplement 3B*). Nonetheless, the range of stimuli over which the critical

networks show pre-stimulus amplitude regulation is superior to that of the subcritical and supercritical networks, regardless of the strength of the input weight (*Figure 2—figure supplement 3C*).

## Pre-stimulation phase influence on phase-locking requires critical dynamics

Having identified a strong interaction between criticality of oscillations and pre-stimulus amplitude on stimulus-evoked phase-locking, we asked whether a similar effect is present for the phase of the pre-stimulus oscillation. For the sub- and super-critical networks there was no pre-stimulus phase regulation of response at a stimulus size of 5 neurons (*Figure 3A*). In contrast, the critical network showed large differences in post-stimulus phase-locking for different pre-stimulus phase bins (*Figure 3A*). These networks showed the weakest response on the descending side of the alpha cycle, when the network is inhibited, and the strongest response on the rising side of the alpha cycle, when the network is in an excitable state (*Figure 3—figure supplement 1*). Looking across the connectivity parameter space, we found that pre-stimulus phase dependence of post-stimulus phase-locking requires a balance in E/I connectivity (*Figure 3B*), and is significantly correlated with criticality as reflected in LRTC (*Figure 3C*, *Figure 3—figure supplement 2*). The pre-stimulus phase influence on phase-locking also generalized to other stimulus sizes. Given the three combinations of excitatory/inhibitory connectivity indicated in *Figure 3D*, we found that the critical networks show pre-stimulus regulation over the widest range of stimulus sizes. We found that changing the weight of the external input also affects the range of stimuli over which pre-stimulus phase regulation is

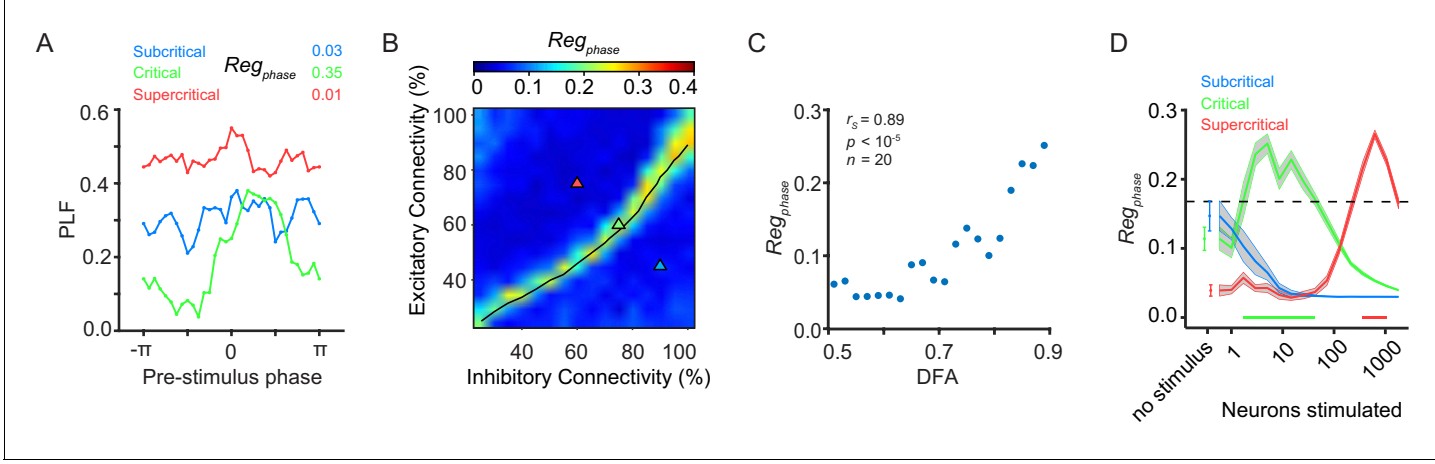

**Figure 3.** Pre-stimulus phase regulation of response requires critical-state dynamics. (A) The phase of pre-stimulus oscillations can alter the phase-locking response of a network to a stimulus in the presence of critical oscillations. Trials are split into 32 evenly spaced bins based on pre-stimulus phase (−5 ms) with an equal number of trials in each bin. Phase-locking at 150 ms post-stimulus was calculated for each bin. Pre-stimulus phase regulation of response, $Reg_{phase}$, was calculated based on the post-stimulus PLF distribution of the bins and examples are shown for sub-critical (*blue*), critical (*green*) and super-critical (*red*) networks. Super-critical networks show higher PLF per bin than sub-critical networks because the former have stronger alpha oscillations which carry the phase of alpha oscillations for a longer time into the post-stimulus period without adjusting the phase due to the stimulus. (B) Pre-stimulus phase regulation is dependent on E/I connectivity balance. The *black line* indicates critical neuronal avalanches, as measured when the networks were not stimulated. Here, shown for a stimulus size of 5 neurons. (C) Pre-stimulus phase regulation is more prominent for stronger LRTC at 150 ms post-stimulus (Spearman correlation). DFA was computed for the unstimulated networks, and $Reg_{phase}$ for the same networks, but with five neurons stimulated. Networks binned based on DFA exponent, as in *Figure 1F*. (D) Critical networks exhibit pre-stimulus phase regulation over the widest range of stimulus sizes. Supercritical networks also show regulation, but only at very high stimulus sizes. mean ± sem corresponding to 10 individual networks with the same excitatory/inhibitory connectivity as the selected sub-critical (*blue*), critical (*green*), super-critical (*red*) networks. Black shaded line is the significance threshold calculated as the 95% percentile of pre-stimulus phase regulation based on bootstrapping of pre-stimulus and post-stimulus phases (see Supplementary information - Materials and methods).

The online version of this article includes the following figure supplement(s) for figure 3:

**Figure supplement 1.** Critical networks show maximal phase-locking response on the rising side of the alpha oscillation, and minimal phase-locking response on the falling side of the alpha oscillation.

**Figure supplement 2.** Critical networks exhibit maximum phase regulation at multiple post-stimulus latencies.

**Figure supplement 3.** Critical networks exhibit maximum phase regulation for a wide range of input-stimulus strengths.

significant (*Figure 3—figure supplement 3B*). Nonetheless, pre-stimulus phase-regulation occurs for a wider range of stimuli in critical networks than in their subcritical or supercritical counterparts, regardless of the strength of the input weight (*Figure 3—figure supplement 3C*).

## The relationship between criticality and network versatility is robust to changes in model parameters

Model parameters were selected through an optimization process which aimed at getting power-law avalanche distributions and robust LRTC (as described in more detail in Materials and methods), but did not include any criteria to maximize the dynamic range of phase-locking response, or pre-stimulus amplitude or phase regulation. Therefore, we expect that the functional consequences of criticality on stimulus processing that emerged out of the model, are generic to criticality, and not specific to the actual parameters. To verify this, we repeated our analysis for two versions of the model where the synaptic weight of excitatory to excitatory connections ($w_{EE}$) was multiplied by a factor of 0.75, or by a factor of 1.25. Multiplication of $w_{EE}$ by a factor of 0.75 reduced the overall activity in the network, which means that stronger connectivity is required to achieve the same dynamics, leading to a shift up of the critical line through the phase space (*Figure 4*). Increasing $w_{EE}$ had the opposite effect on the critical line. Importantly, the functional consequences of criticality—maximized dynamic range, pre-stimulus amplitude and phase regulation—followed this shift in the

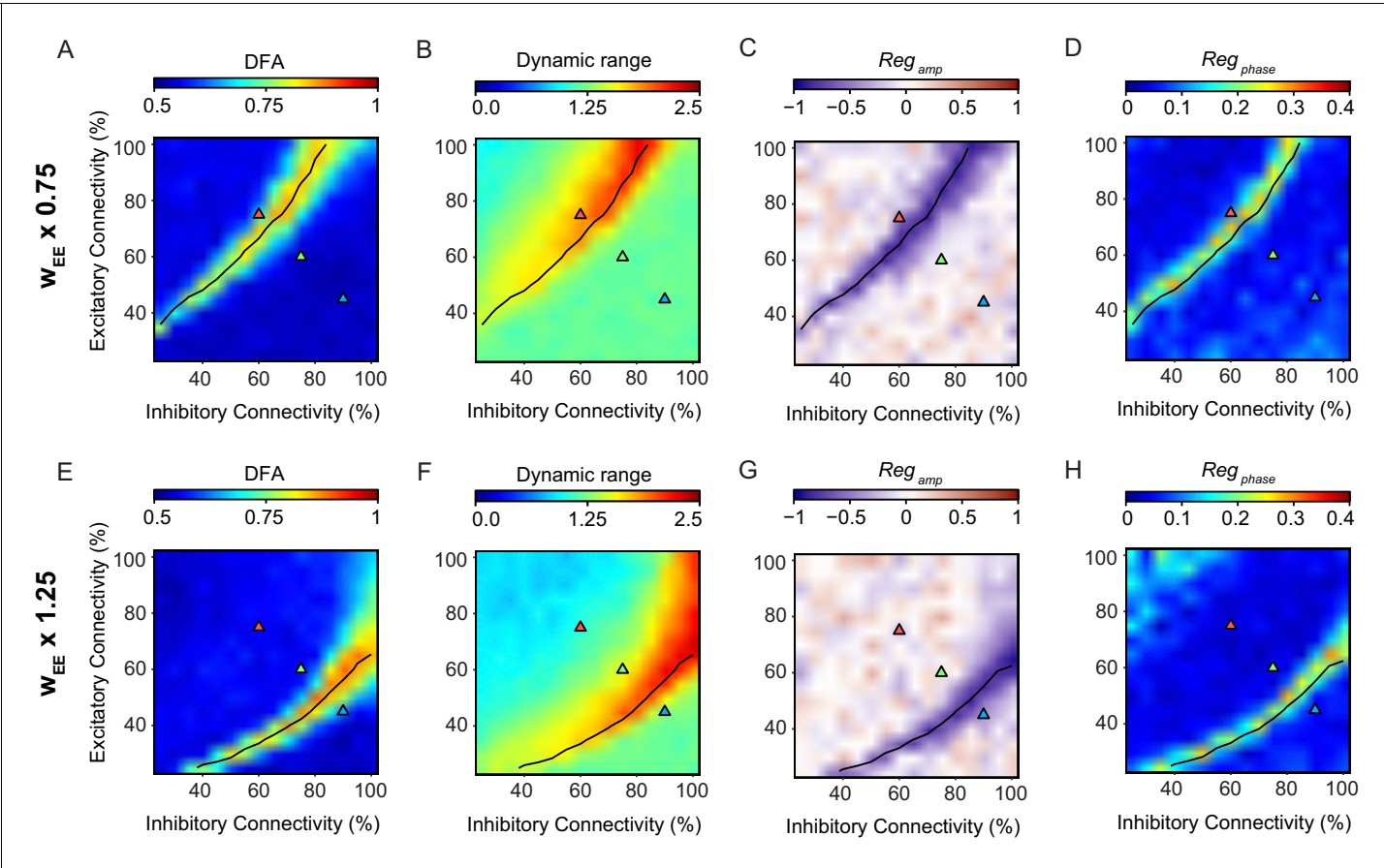

**Figure 4.** The relationship between criticality and network versatility is robust to changes in the strength of excitatory connections. Phase-space plots of LRTC (A,E), Dynamic Range (B,F) Amplitude Dependence (C,G), and Phase Dependence (D,H), for networks with the weight of the excitatory to excitatory connections ($w_{EE}$) multiplied by a factor of 0.75 (top row) or 1.25 (bottom row). *black line* indicates critical neuronal avalanches, as measured when the networks were not stimulated.

critical line. This suggests that the relationship between criticality and versatility in stimulus processing is not coincidental on the specific choice of parameters, but generic to criticality.

## Discussion

In this study, we establish for the first time a link between critical-state dynamics of oscillations and the ability of a network to respond differently depending on the amplitude of ongoing alpha oscillations. This functionality has been investigated in many human studies (*Ergenoglu et al., 2004*; *Hanslmayr et al., 2005*; *Linkenkaer-Hansen et al., 2004*), where a low oscillation power in relevant sensory areas in the pre-stimulus period has been seen as a pre-requisite of conscious perception. In non-relevant areas, on the other hand, it has been suggested that a high oscillation power serves to functionally inhibit that region (*Smith et al., 2012*). Our findings are also consistent with the notion that alpha oscillations create periodic fluctuations in excitability (*Klimesch et al., 2007*), which are indexed by the phase of the oscillations. Pre-stimulus phase has been shown to bias perception (*Busch et al., 2009*; *Mathewson et al., 2009*; *VanRullen, 2016*) and we found that this function is also likely to be modulated by the criticality of the network. In addition, we show that oscillatory spiking neuronal networks have an optimal dynamic range when they are critical, which is in agreement with previous work on non-oscillatory networks (*Gautam et al., 2015*; *Kinouchi and Copelli, 2006*; *Larremore et al., 2011*; *Shew et al., 2009*), and suggests that optimal dynamic range is a generic property of critical systems. Thus, since the level of criticality affects network response in terms of stimulus phase-locking, and phase-locking plays a role in the generation of event-related responses (ERPs) (*Makeig et al., 2002*; *Min et al., 2007*), we predict that criticality will also affect the different time components of ERPs.

There is much debate as to the cause of amplitude fluctuations during threshold stimulus detection tasks, with top-down mechanisms such as attention and expectation being suggested (*Jensen et al., 2012*; *Melloni et al., 2011*). Here, we show that a network with balanced excitatory and inhibitory connectivity produces spontaneous amplitude fluctuations that have a functional role, meaning that the network goes through periods of high and low sensitivity to external stimuli. In the unbalanced state, the network loses this function and either reliably reacts (in the sub-critical case), or ignores the stimulus (super-critical case). Thus, top-down mechanisms are not required for fluctuations in alpha power; however, it is plausible that attentional mechanisms can actively regulate the level of fluctuations to shift the network's operating point relative to the critical state, allowing the network to react in the required manner (*Wyart and Tallon-Baudry, 2009*). How do these findings relate to task performance? Pre-stimulus activity regulation of post-stimulus response has been found beneficial, by enhancing detection of stimuli that are attended to (*Capilla et al., 2014*) or expected (*Mayer et al., 2016*), and reducing detection of irrelevant stimuli. The spontaneous fluctuations in amplitude observed at criticality, with impact on conscious perception, may serve to flexibly switch between the different stimuli competing for attention. To this is added another advantage of being in the critical state: it requires minimal attentional modulations to reach the sub- or super-critical states, and thus to enhance aspects of stimuli, for example in the case of stimulus discrimination (*Tomen et al., 2014*).

However, is this network ability to regulate perception based on pre-stimulus amplitude always optimal for performing a task? In tasks where continuous detection ability is required, fluctuations in pre-stimulus activity will actually decrease task performance in those periods of amplitude that produce smaller post-stimulus response. In addition, the broadening of attention to external stimuli in the critical state, is accompanied by increased susceptibility to task-unrelated mentation, or mind-wandering (*Hellyer et al., 2014*). When people experience periods of mind-wandering during task-performance, this results in reduced phase-locking to a stimulus (*Baird et al., 2014*). In these cases a more sub-critical network would actually perform better in terms of constant attention (*Irrmischer et al., 2018*; *Tomen et al., 2014*) and a super-critical network would perform better in terms of constant functional inhibition. Our results are in line with the predictions of *Cramer et al. (2019)*, which suggest that the critical state is not optimal for all information processing requirements: complex tasks benefit from criticality, whereas in simple tasks, the high susceptibility to stimuli, memory capacity and entropy of a critical regime might impair performance.

We have shown that critical networks have overall the widest dynamic range of phase-locking response, and pre-stimulus amplitude and phase regulation. However, we also found that

supercritical networks can show differential PLF and pre-stimulus regulation for the large stimulus sizes where the PLF response of critical networks saturates. One way in which the brain can take advantage of the information processing benefits of different states is to regulate its dynamics with respect to the critical point in a task-dependent manner (*Pfeffer et al., 2018*). *Zierenberg et al. (2020)* show that the brain could also combine networks with different distances to the critical point, such that the resulting ensemble has a dynamic range of stimulus response wider than any of the ensemble networks. This strategy could plausibly be employed to extend the range of stimulus sizes over which pre-stimulus amplitude and phase regulation occur.

Although our computational model abstracts many biophysiological neuronal mechanisms, it captures essential features of neuronal dynamics: alpha oscillations, long-range temporal correlations, and scale-free avalanches, which emerge at a specific balance of excitatory and inhibitory connectivity. We showed that the relationship between criticality and the maximal versatility of network function is not specific to the selection of model parameters, which suggests that it is generic to criticality. Future studies should test whether our results also hold for more physiologically realistic models, for example where excitatory and inhibitory currents balance each other (*Haider et al., 2006*; *Rudolph and Destexhe, 2003*).

Previously, non-oscillatory neuronal models have shown maximized dynamic range at criticality (*Gautam et al., 2015*; *Kinouchi and Copelli, 2006*; *Larremore et al., 2011*; *Shew et al., 2009*). While it is plausible that non-oscillatory models could also show a pre-stimulus 'amplitude' regulation effect based on the level of pre-stimulus firing—and that this effect is likely to be maximal at criticality—it is unclear how a phase regulation effect could occur without an ongoing oscillation. Future studies could investigate whether the presence of oscillations is important for pre-stimulus phase and amplitude regulation to occur, in models with a different type of critical phase transition (such as absorbing/sustained) (*Del Papa et al., 2017*; *Levina et al., 2007*).

Overall our study highlights the importance of critical-state dynamics (*Chialvo, 2010*) for understanding neuronal network function, and shows that it is possible to integrate oscillatory mechanisms into this framework. Importantly, while the quantitative hallmarks of criticality require long-time monitoring of spatial and temporal dynamics that are statistically stable, our results show that the concept of critical brain dynamics is compatible with time-varying functions—an important notion in contemporary theories of neuronal oscillations and their role in neuronal communication and in attention.

# Materials and methods

## Key resources table

| Reagent type (species) or resource | Designation | Source or reference | Identifiers |
|---|---|---|---|
| Software, algorithm | Brian Simulator | https://briansimulator.org | RRID:SCR_002998 |
| Software, algorithm | MATLAB | https://www.mathworks.com/products/matlab | RRID:SCR_001622 |
| Software, algorithm | circular statistics | https://mathworks.com/matlabcentral/fileexchange/10676-circular-statistics-toolbox-directional-statistics | RRID:SCR_016651 |
| Software, algorithm | R Project for Statistical Computing | http://www.r-project.org/ | RRID:SCR_001905 |

## CRitical OScillations (CROS) model

We used an adapted version of the model described in *Poil et al. (2012)*, with synaptic weights optimized for power-law avalanches and long-range temporal correlations. The model was implemented using the Brian2 simulator for spiking neural networks (*Stimberg et al., 2014*; RRID:SCR_002998). Specifically, we modeled networks of 75% excitatory and 25% inhibitory integrate-and-fire neurons arranged on a 50x50 open grid. Placing neurons on the grid using uniform sampling may result, by chance, in clusters of excitatory or inhibitory neurons. To avoid this, we first positioned inhibitory neurons on the grid according to Mitchell's best candidate algorithm (*Mitchell, 1991*), which results

in a more even spatial distribution. The remaining spaces were filled with excitatory neurons. Networks differ in their two connectivity parameters, $C_E$ and $C_I$, which are the percentage of other neurons within a local range (a circle with radius = 4 neurons centered on the presynaptic neuron) that each excitatory and each inhibitory neuron connects to, respectively. Connectivity parameters were set between 25 and 100% at 5% intervals, and 10 different networks were created for each combination of $C_E$ and $C_I$. Border neurons had fewer connections because these neurons had a lower number of neurons in their local range. Within local range, connection probability decreases exponentially with distance. More specifically, the probability, $P$, of a connection at a distance $r$ was given by:

$$P(r) = \min(\alpha e^{-r}, 1) \qquad (1)$$

where $\alpha$ is optimized separately for excitatory and inhibitory neurons such that the overall connectivity probability within a neuron's local range is equal to $C_E$ or $C_I$, depending on whether the neuron is excitatory or inhibitory. For example in the case of an excitatory neuron $i$, with connectivity probability $C_E$, where the set of neighboring neurons within the local range is $J$, and $|J|$ is the number of neighbors, $\alpha$ will have to satisfy the following equation:

$$\sum_{j \in J} P(r_j) = \sum_{j \in J} \min(\alpha e^{-r_j}, 1) = C_E |J| \qquad (2)$$

As such, we used the Nelder-Mead optimization algorithm to determine the value of $\alpha$ which minimizes the following function:

$$f(\alpha) = \left| \sum_{j \in J} \min(\alpha e^{-r_j}, 1) - C_E |J| \right| \qquad (3)$$

## Neuron model

Neurons were modeled using a synaptic model integrating received spikes, and a probabilistic spiking model. Each time step ($dt$) of 1 ms starts with each neuron, $i$, updating the input $I_i$ with spikes received from the presynaptic neurons $J_i$, together with an exponential synaptic decay:

$$I_i(t + dt) = \left( I_i(t) + \sum_{j}^{J_i} W_{ij} S_j(t) \right) \left( 1 - \frac{dt}{\tau_I} \right) \qquad (4)$$

The weights $W_{ij}$ are fixed, and depend on the type of the pre- and post-synaptic neuron. $\tau_I$ is the decay constant of inputs, and $S$ is a is a binary vector, with $S_j=1$ if the pre-synaptic neuron $j$ fired in the previous time step, and $S_j=0$ otherwise.

The activation of a neuron $A_i$, is then updated with these excitatory and inhibitory inputs, together with an exponential decay, $\tau_P$, and a baseline activation level $A_0$:

$$A_i(t + dt) = (A_i(t) + I_i(t)) \left( 1 - \frac{dt}{\tau_P} \right) + A_0 \frac{dt}{\tau_P} \qquad (5)$$

The spiking probability $P_i^s$ is calculated as a function of the neuron activation $A$ at the current timestep, as follows:

$$P_i^s(t) = \begin{cases} 0, & A_i(t) < 0 \\ A_i(t), & 0 \leq A_i(t) \leq 1 \\ 1, & A_i(t) > 1 \end{cases} \qquad (6)$$

We determine whether the neuron spikes with the probability $P_S$. If a neuron spikes, the neuron activation $A$ is reset to the reset value, $A_r$. At the next time step, all neurons that it connects to will have their input updated again according to *Equation 4*.

## Model parameters

All model parameters were the same as in the original paper (*Poil et al., 2012*) apart from the synaptic weights. Neuron model: ($\tau_I$ = 9 ms), Synaptic model: Excitatory neurons ($\tau_P$ = 6 ms, $A_0$ = 0.000001, $A_r$ = -2), Inhibitory neurons ($\tau_P$ = 12 ms, $A_0$ =0, $A_r$ = -20). To improve the range and

stability of the long-range temporal correlations from the original model, an evolutionary algorithm (*Smit and Eiben, 2011*) was applied to the synaptic weights. The parameters that could vary were the 2 connectivity parameters (taking values between 0 and 100%), and the natural logarithm of the magnitude of the 4 synaptic weights, $W_{EE}$, $W_{IE}$, $W_{EI}$ and $W_{II}$ (taking values between -5 and 1). For each run, a fitness value was calculated based on the avalanches size and duration distributions (see Materials and methods - Neuronal Avalanches), and the LRTCs (see Materials and methods - Detrended fluctuation analysis of long-range temporal correlations).

$$fitness = \frac{1}{|1-DFA| + |1-\kappa_{size}| + |1-\kappa_{duration}|} \tag{7}$$

The optimum weights ($W_{ij}$, connecting the presynaptic neuron $j$ to the postsynaptic neuron $i$) found by the algorithm were ($W_{EE} = 0.0085, W_{IE} = 0.0085, W_{EI} = -0.569, W_{II} = -2$).

## Network activity analysis

A network signal was created by summing the total number of neurons spiking at each time-step with a Gaussian white noise signal of the same length with mean = 0 and σ = 3. This level of white noise was set to allow all networks to achieve a time-varying phase, which is not the case without adding the noise, when there are silent periods in the network. Detrended fluctuation analysis, oscillation power, and phase-locking factor, which are described below, were calculated on this pre-processed signal, with Gaussian noise added. The analysis of neuronal avalanches was performed on the raw signal, consisting of the total number of neurons spiking at each time step.

## Oscillation power

The power spectrum was computed using the Welch method with a Hamming window with $2^{11}$ FFT points.

## Neuronal avalanches

A neuronal avalanche is defined as a period where neurons are spiking above a certain threshold—in our case set to half median of activity. The size of the avalanche is the number of spikes during this period. We then computed the κ index (*Poil et al., 2012*; *Shew et al., 2009*), which determines the difference between the distribution of our data and a power-law, by calculating the average difference of the cumulative distribution of a power-law function, $P$, (with exponent −1.5 for size and −2.0 for duration) and that of our experimental data, $A$, at 10 equally spaced points on a logarithmic axis ($\beta$) and adding 1.

$$\kappa = \frac{1}{10} \sum_{i=1}^{10} (P(\beta_i) - A(\beta_i)) + 1 \tag{8}$$

A subcritical distribution is characterized by κ <1, and a supercritical distribution by κ >1, whereas κ = 1 indicates a critical network.

## Detrended fluctuation analysis of long-range temporal correlations

Detrended fluctuation analysis (DFA) was used to analyze the scale-free decay of temporal (auto)correlations, also known as long-range temporal correlations (LRTC). The DFA was introduced as a method to quantify correlations in complex data with less strict assumptions about the stationarity of the signal than the classical autocorrelation function or power spectral density (*Hardstone et al., 2012*; *Linkenkaer-Hansen et al., 2001*). In addition, DFA facilitates a reliable analysis of LRTC up to time scales of at least 10% of the duration of the signal (*Chen et al., 2002*; *Gao et al., 2006*). DFA exponents in the interval of 0.5 to 1.0 indicate scale-free temporal correlations (autocorrelations), whereas an exponent of 0.5 characterizes an uncorrelated signal. The main steps from the broadband signal to the quantification of LRTC using DFA have been explained in detail previously (*Hardstone et al., 2012*; *Linkenkaer-Hansen et al., 2001*). In brief, the DFA measures the power-law scaling of the root-mean-square fluctuation of the integrated and linearly detrended signals, $F(t)$, as a function of time-window size, $t$ (with an overlap of 50% between windows). The DFA exponent is the slope of the fluctuation function $F(t)$, and can be interpreted as the strength of the

autocorrelations in signals. For our analyses we computed DFA on the amplitude envelope of the signal filtered in the 8–16 Hz range, and the exponent was fit between 2 and 50 s.

## Unstimulated networks

To assess the dynamics of unstimulated networks we allowed connectivity to take values between 25 and 100% at 5% intervals. For all 256 possible parameter combinations of excitatory and inhibitory connectivity, we created 10 different networks and ran each network for $2 \times 10^6$ time-steps (2000 s).

## Stimulation networks

To test network response to a stimulus we took one sample network for each combination of excitatory and inhibitory connectivity parameters, and for each run of the network we stimulated $n$ excitatory neurons where $n \, \varepsilon$ {1, 2, 3, 5, 9, 15, 25, 43, 74, 126, 216, 369, 632, 1081, 1800}. The stimulated neurons were randomly distributed across the grid. For all 256 possible parameter combinations of excitatory and inhibitory connectivity, and stimulus size, we used the same 10 different networks as in the unstimulated case, and ran each network for $2 \times 10^6$ time-steps (2000 s). During the stimulation runs all neurons in the receptive field simultaneously received a stimulus once in every 500–1500 ms. The stimulus had the same weight as any of the excitatory-excitatory connections in the network ($w_{input} = w_{EE} = 0.0085$), and was applied simultaneously, for one timestep = 1 ms, to all the $n$ excitatory neurons which the stimulus connected to. Additionally, we investigated the effect of the stimulus strength on network dynamics by modulating $w_{input}$ by a factor of 1/64 to 64 times the original value, on a log2 scale.

## Phase-locking factor

Stimulus response was calculated in the network in terms of the phase-locking factor (*Palva et al., 2005*). Data were filtered using a one-way causal FIR filter (order 250) between 8 and 16 Hz. The use of a causal FIR filter ensured that pre-stimulus phase and amplitude were not contaminated by the stimulus responses, at the cost of delaying the peak response to the stimulus by 125 ms. The phase at any point in time was obtained by taking the angle of the Hilbert transform. Phase-locking factor (PLF) measures the uniformity of phases across trials at a time-point post-stimulus and was calculated as the mean vector length of all phases (*Berens, 2009*; RRID:SCR_016651):

$$\mathrm{PLF} = \left\| \frac{1}{\mathrm{N}} \sum_i^T z_i \right\| \tag{9}$$

where $T$ is the set of trials, of size $N$, and $z_i$ is the unit vector whose angle corresponds to the momentary phase in trial $i$, at the time point for which PLF is calculated.

## Dynamic range

We fit a sigmoid to the PLF response at 150 ms post-stimulus, as a function of the number of neurons stimulated. The dynamic range is calculated as the orders of magnitude of neurons stimulated between the 10% and 90% of the sigmoid we fit (*Kinouchi and Copelli, 2006*).

## Pre-stimulus amplitude regulation of PLF

To calculate the PLF response corresponding to different levels of pre-stimulus amplitude, we separated trials into 10 percentile bins, based on the average amplitude in the −150 to −50 ms time-range, calculated for the 8–16 Hz frequency band. The percentile binning procedure ensured an equal number of trials across bins, which is important because the PLF value reflects the distribution of phases, but is also determined by the number of trials (e.g., the PLF value for uniformly randomly distributed phases is dependent on the number of trials). PLF at 150 ms post-stimulus was calculated separately for each of the 10 bins. The regulation measure $Reg_{amp}$ was calculated as the Spearman correlation between the index of the bin and the PLF for that bin.

## Pre-stimulus phase regulation of PLF

To calculate the pre-stimulus phase regulation of PLF, trials were split into 32 overlapping bins of width π/8 based on the pre-stimulus phase in the 8–16 Hz frequency range at time −5 ms. To get an equal number of trials in each bin, $x$ randomly selected trials from each bin were picked where $x$ was the smallest number of trials in a bin. PLF at 150 ms post-stimulus was calculated separately for each bin. If no phase regulation takes place, a uniform distribution of post-stimulus PLF is expected. The calculation of phase regulation uses the phase-locking factor of each of 32 phase bins as the weight for the unit vector of that phase. The sum of the 32 vectors is then normalized by the sum of all PLFs. For this calculation, we used the circ_r function from the circ_stats package (*Berens, 2009*; RRID:SCR_016651):

$$Reg_{phase} = \left\| \frac{1}{\sum\limits_{i}^{P} PLF_i} \sum\limits_{i}^{P} PLF_i z_i \right\| \tag{10}$$

where $P$ is the set of 32 bins, $z_i$ is the unit vector whose angle corresponds to the phase at the center of each bin $i$, and $PLF_i$ is the phase-locking factor across all trials belonging to bin $i$. This measure gives a value of $Reg_{phase}$ between 0 (PLF is same for all bins) and 1 (PLF is non-zero in one bin, and zero in all others).

## Association between criticality and network function

Criticality was measured by the LRTC (DFA) of alpha oscillations in the non-stimulated networks. Network function was measured by pre-stimulus amplitude and phase modulation, as well as dynamic range, of the stimulated networks. To get robust estimators for criticality as well as our 3 indicators of network function, we averaged the values across all 10 network initializations, for each excitatory/inhibitory connectivity parameter combination ($n$=256). Before correlating criticality to network function, we had to address another aspect: a disproportionate amount of networks had low DFA. To this end, we binned the $n$=256 datapoints corresponding to each aspect of network function (pre-stimulus amplitude and phase modulation, dynamic range), by DFA, in bins of size 0.02, covering a DFA range of 0.5 to 0.9. The association between criticality and pre-stimulus amplitude regulation, for example, was computed as the Spearman correlation between the mean values of the 20 DFA bins, and the mean $Reg_{amp}$ for the networks belonging to that bin. The same type of pre-processing was performed for pre-stimulus phase regulation and dynamic range. To identify the time ranges where criticality predicts network function, we calculated the Spearman correlation mentioned earlier, for all timepoints between 0 and 500 ms post-stimulus. To correct for multiple comparisons, we used the cluster-based correction with permutation (*Nichols and Holmes, 2002*), with a significance threshold $\alpha = 0.001$. We will exemplify again the process by looking at the association between criticality and pre-stimulus amplitude regulation, but the steps are the same for the other indicators of network function. First, we calculated the cluster significance threshold, to determine which clusters could have arisen by chance. While keeping the ordering of the DFA bins the same, we permuted the corresponding $Reg_{amp}$ values, separately for each timepoint between 0 and 500 ms post-stimulus. We then computed the Spearman correlation coefficients $r_S$ between DFA and the permuted $Reg_{amp}$, for each timepoint, and identified all clusters – sets of contiguous timepoints with a p-value for the Spearman correlation $p \leq \alpha$. We defined cluster size $S_c = \sum\limits_{t}^{T_c} |r_S^t|$, where $t$ is a time-point belonging to the set of contiguous timepoints $T_c$ in cluster $c$, and $|r_S^t|$ is the absolute value of the Spearman correlation coefficient at time $t$. We then calculated the maximum cluster size $S_{max_i}$ for the permuted dataset $i$ to be $\max\limits_{c \in C} S_c$, where $C$ is the set of all clusters $C$. We repeated this process 10000 times, which gave us a distribution of $S_{max_i}$ over different permutations of the data. We set the cluster significance threshold $T_{S_{max}}$ to correspond to the $(1 - \alpha) * 100$ percentile of $S_{max_i}$. We then identified all clusters from the actual (non-permuted data), consisting of contiguous timepoints with with a p-value for the Spearman correlation $p \leq \alpha$, and kept only those clusters, whose cluster size $S_c > T_{S_{max}}$.

## Baseline level of pre-stimulus regulation

For this, we generated 10 synthetic runs (equal to the number of simulations of the CROS model for each excitatory/inhibitory connectivity combination)—each consisting of 2000 pairs of pre/post stimulus phase (equal to the number of times that the CROS model was stimulated throughout a run) drawn from a uniform distribution, and calculated pre-stimulus phase regulation for each of the 10 synthetic runs. Then we computed the average phase regulation across all 10 runs. We repeated this process 10000 times to get the 95% percentile of phase regulation. We used this as a significance threshold to determine whether 10 individual subcritical, critical, or supercritical networks, at a certain stimulus size, showed pre-stimulus phase regulation above what is expected from a random distribution of pre/post-stimulus phases. As seen in *Figure 3D*, sub-critical networks at strong stimuli show phase regulation well below this threshold. That is because these networks respond reliably to the strong stimuli regardless of the pre-stimulus phase, thus the distribution of responses across phases is more uniform than expected from a random distribution. Some supercritical networks show phase regulation below threshold for the weaker stimuli, because they have strong alpha oscillations which maintain the phase of oscillations similar for a long time after they have been binned uniform.

## Robustness of relationship between network versatility and criticality

To verify that our results are not dependent on the specific model parameters, we used two variants of the model, where the weight of excitatory to excitatory connections ($w_{EE}$) was multiplied by a factor of 0.75 and 1.25, respectively. For each combination of excitatory and inhibitory connectivity, we created five different networks. We then ran each network for $2 \times 10^6$ time-steps (2000 s), for all combinations of stimulus size and weight of excitatory to excitatory connections ($w_{EE}$).

## Acknowledgements

This work was funded by Netherlands Organization for Scientific Research (NWO) Physical Sciences Grant 612.001.123 (to KL-H and RH) and NWO Social Sciences grant 406.15.256 (to KL-H and A-EA). We thank Simon-Shlomo Poil for discussions on the manuscript.

## Additional information

### Funding

| Funder | Grant reference number | Author |
|---|---|---|
| Nederlandse Organisatie voor Wetenschappelijk Onderzoek | 612.001.123 | Richard Hardstone<br>Klaus Linkenkaer-Hansen |
| Nederlandse Organisatie voor Wetenschappelijk Onderzoek | 406.15.256 | Arthur-Ervin Avramiea<br>Klaus Linkenkaer-Hansen |

The funders had no role in study design, data collection and interpretation, or the decision to submit the work for publication.

### Author contributions

Arthur-Ervin Avramiea, Software, Formal analysis, Validation, Investigation, Visualization, Methodology; Richard Hardstone, Jan-Matthis Lueckmann, Conceptualization, Software, Formal analysis, Validation, Investigation, Visualization, Methodology; Jan Bím, Software, Methodology; Huibert D Mansvelder, Conceptualization, Supervision; Klaus Linkenkaer-Hansen, Conceptualization, Supervision, Funding acquisition, Methodology, Project administration

### Author ORCIDs

Arthur-Ervin Avramiea https://orcid.org/0000-0002-0826-8269
Richard Hardstone https://orcid.org/0000-0002-7502-9145
Jan-Matthis Lueckmann https://orcid.org/0000-0003-4320-4663
Jan Bím https://orcid.org/0000-0003-2780-5610

Huibert D Mansvelder (iD) https://orcid.org/0000-0003-1365-5340
Klaus Linkenkaer-Hansen (iD) https://orcid.org/0000-0003-2140-9780

### Decision letter and Author response
Decision letter https://doi.org/10.7554/eLife.53016.sa1
Author response https://doi.org/10.7554/eLife.53016.sa2

## Additional files

### Supplementary files
• Transparent reporting form

### Data availability
Source code required to run all simulations, as well as datasets and scripts required to generate all figures presented here, are available at https://doi.org/10.6084/m9.figshare.12167943.v2.

The following dataset was generated:

| Author(s) | Year | Dataset title | Dataset URL | Database and Identifier |
|---|---|---|---|---|
| Avramiea A-E, Hardstone R, Lueckmann J-M, Bím J, Mansvelder HD, Linkenkaer-Hansen K | 2020 | Pre-stimulus phase and amplitude regulation of phase-locked responses is maximized in the critical state | https://doi.org/10.6084/m9.figshare.12167943.v2 | figshare, 10.6084/m9.figshare.12167943.v2 |

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
