## [Decision Letter]

**Acceptance summary:**

There has been much interest in how stimulus processing depends on the phase and amplitude of pre-stimulus oscillatory activity. Using sophisticated computational modeling, this paper demonstrates that near-criticality is required to enable the ability of networks to regulate stimulus response based on pre-stimulus activity. This highlights a potential functional role for the critical state in allowing pre-stimulus states to influence sensory information processing.

**Decision letter after peer review:**

Thank you for submitting your article "Versatility of neuronal network function is maximized in the critical state" for consideration by *eLife*. Your article has been reviewed by two peer reviewers, and the evaluation has been overseen by Floris de Lange as the Senior Editor and Reviewing Editor. The reviewers have opted to remain anonymous.

Summary:

The authors study the stimulus-evoked responses in a previously published model (from the same group), which generates alpha oscillations and long-range temporal correlations. Previous results have shown that this model can exhibit critical dynamics depending on the connection probabilities of excitation and inhibition. The main novel contribution of this work is that it shows that the pre-stimulus biases (alpha frequency amplitude and phase) of the stimulus-evoked responses only occur in networks close to criticality. In particular, they found a strong negative correlation between pre-stimulus alpha amplitude and phase-locking response to stimulus in critical networks. This is consistent with experimental results that show higher alpha activity is correlated with decreased stimulus detection. This result also suggests that attention can modulate the network from the supercritical state where it ignores stimulus to the subcritical state where it reliably responds to stimulus.

In addition, the authors also show that networks at critical state have the largest dynamic range of phase-locking responses to stimulus. However, similar result has been shown in previous work for the dynamic range of firing rate responses.

Essential revisions:

1) Physiological realism.

Please report how the firing rates of excitatory and inhibitory neurons depend on E-I ratio (analog figure to e.g. 1E, but displaying the firing rates).

Please show raster plots of spiking for e.g. 100 randomly chosen neurons; a 250-500 ms window would be good to see; in addition, show population rates for a few seconds; both for the three examples (sub/super/critical).

"E-I balance" in the strict sense refers to excitatory and inhibitory currents within a single neuron canceling each other. Does the model show that?

2) Robustness.

You modified parameters with respect to the first version of the model. Ideally, the effects you show do not depend on these parameter modifications. Could you demonstrate that?

Likewise, would you expect that other models, like Levina, Hermann and Geisel, 2007, or del Papa, Priesemann and Triesch, 2017, would show similar types of effects, even in the absence of oscillations, thus at the transition between absorbing/sustained activity?

3) Dynamic Range.

The dynamic range at criticality is larger than at sub/supercritical models. However, the "working point" or response range also changes considerably. This is in analogy to Zierenberg et al., 2020. How do your results relate on the systematic change of the response range, and to their ideas that combining networks of different distances to criticality would greatly enhance the dynamic range?

4) Writing.

As to the writing, I think this paper is too concise and requires prior knowledge of the previously published papers from this group. I would appreciate more explanations in the main text about the key terminologies and concepts, such as "critical oscillation", "Long-range temporal correlations", "Detrended fluctuation analysis".

5) Data and code sharing.

The raw spike data for the three states (sub/super/critical) should be made available, so that everyone can re-analyse them. The code should be made available as well. Please share the code before acceptance.

6) I didn't find a clear definition of the stimulus input? What's the duration and magnitude? Is it a constant current in time? Why is the stimulus strength defined as the number of stimulated neurons? Would results be similar if stimulus strength is modeled as the magnitude of input current?

7) Figure 1D, if input current is larger, does it expand the dynamic range of the supercritical networks?

8) Figure 1C, Is the phase-locking response related to the onset increase of firing rate? If the network shows onset response in rate, it would result in more rising phases, right? Since previous work has shown criticality maximizes dynamic range of evoked rate responses, how is this result different from previous work?

9) Figure 3D, Similar to my comments above about the dynamics range, I wonder if the range of pre-stimulus regulation depends on the magnitude of input current. If you drive the supercritical network with stronger input for each neuron, does it expand the region of significant regulation.

10) About the balance between excitation and inhibition: I think it should be the product of synaptic weight and connection probability that determines the network dynamics. Why is the critical transition at approximately when excitatory connectivity percentage equals inhibitory connectivity percentage? How does the transition depend on the recurrent weights?

---

## [Author Response]

Essential revisions:1) Physiological realism.Please report how the firing rates of excitatory and inhibitory neurons depend on E-I ratio (analog figure to e.g. 1E, but displaying the firing rates).

We have added the phase-space plots of excitatory and inhibitory firing rates to Figure 1—figure supplement 1 and refer to the figure in Results:

“The firing rate grew monotonously for both excitatory and inhibitory neurons, as we increased excitatory connectivity (Figure 1—figure supplement 1B,C).”

Please show raster plots of spiking for e.g. 100 randomly chosen neurons; a 250-500 ms window would be good to see; in addition, show population rates for a few seconds; both for the three examples (sub/super/critical).

We agree it is very interesting to illustrate the spiking activity and have added the requested raster plots for stimulated networks in Figure 1 (Figure 1B in the revised manuscript). This also highlights the oscillatory nature of responses in the CROS model (see point #8 below).

We also made raster plots of spontaneous activity in the unstimulated network. We include this figure here for comparison (Author response image 1); however, we judge it too similar to the new Figure 1B to include it in the manuscript. For the sub-critical, critical and super-critical networks, we selected a representative 500-ms window to exemplify spiking for all neurons, as well as a 5-second window to show the evolution of population activity (Author response image 1). The example sub-critical network exhibits very sparse bursts of activity, the super-critical network is characterized by relentless firing, and the critical network gives rise to avalanche patterns with greater variance in size and spread. Due to the recurrent connectivity of the network, inhibitory activity closely follows excitatory activity. However, excitatory activity dominates when the number of excitatory synapses is much larger than the number of inhibitory synapses, as is the case for the supercritical network (Author response image 1).

**Author response image 1. respfig1:** Network time-series and raster plots for unstimulated networks. Time-series and raster plots for an example sub-critical (**A**), critical (**B**), and super-critical (**C**) network. (Top and middle rows) Network time-series for excitatory (red) and inhibitory (blue) neurons. (Bottom) Raster plot of activity from highlighted region in top row.

"E-I balance" in the strict sense refers to excitatory and inhibitory currents within a single neuron canceling each other. Does the model show that?

In our model, we use probabilistic integrate-and-fire neurons (i.e., not a current-based integrate-and-fire). Thus, it is not possible to analyze the relative balance of excitatory and inhibitory currents in these neurons. However, E/I balance can be defined at different levels of neuronal organization and, here, we use a network-level approach based on the number of excitatory and inhibitory connections, and find that for a certain balance of connections, the model produces critical-state dynamics. Note, that if we change the strength of excitatory or inhibitory connections then the ratio of excitatory and inhibitory connections required for critical-state dynamics will also change (See new Figure 4 and the reply to point #2 below).

We have addressed this point in the manuscript (Discussion section):

“Although our computational model abstracts many biophysiological neuronal mechanisms, it captures essential features of neuronal dynamics: alpha oscillations, long-range temporal correlations, and scale-free avalanches, which emerge at a specific balance of excitatory and inhibitory connectivity. We showed that the relationship between criticality and the maximal versatility of network function is not specific to the selection of model parameters, which suggests that it is generic to criticality. Future studies should test whether our results also hold for more physiologically realistic models, e.g., where excitatory and inhibitory currents balance each other (Haider et al., 2006; Rudolph and Destexhe, 2003).”

2) Robustness.You modified parameters with respect to the first version of the model. Ideally, the effects you show do not depend on these parameter modifications. Could you demonstrate that?

The only parameters that were modified compared to the first version of the model were the shape of the local range, which was changed from square to circular, and the synaptic weights, which came out of an optimization process. We have clarified this in the manuscript (Results section):

“Model parameters were selected through an optimization process which aimed at getting power-law avalanche distributions and robust LRTC (as described in more detail in Materials and methods), but did not include any criteria to maximize the dynamic range of phase-locking response, or pre-stimulus amplitude or phase regulation.”

To further support that our results are generic to criticality and not dependent on the specific parameter values, we ran additional simulations and added the following to the Results:

“Therefore, we expect that the functional consequences of criticality on stimulus processing that emerged out of the model, are generic to criticality, and not specific to the actual parameters. To test this, we repeated our analysis for two versions of the model where the synaptic weight of excitatory to excitatory connections (w*_EE_*) was multiplied by a factor of 0.75, or by a factor of 1.25. Multiplication of w*_EE_* by a factor of 0.75 reduced the overall activity in the network, which means that stronger connectivity is required to achieve the same dynamics, leading to an upward shift of the critical line through the phase space (Figure 4). Increasing w*_EE_* had the opposite effect on the critical line. Importantly, the functional consequences of criticality—maximized dynamic range, pre-stimulus amplitude and phase regulation—followed this shift in the critical line. This suggests that the relationship between criticality and versatility in stimulus processing does not require a specific choice of parameters, but is generic to criticality.”

Likewise, would you expect that other models, like Levina, Hermann and Geisel, 2007, or del Papa, Priesemann and Triesch, 2017, would show similar types of effects, even in the absence of oscillations, thus at the transition between absorbing/sustained activity?

We feel that it goes beyond the scope of the present paper to implement other models in order to answer this question with certainty; however, we have added a comment to the manuscript (Discussion):

“Previously, non-oscillatory neuronal models have shown maximized dynamic range at criticality (Shew et al., 2009). While it is plausible that non-oscillatory models could also show a pre-stimulus “amplitude” regulation effect based on the level of pre-stimulus firing—and that this effect is likely to be maximal at criticality—it is unclear how a phase regulation effect could occur without an ongoing oscillation. Future studies could investigate whether the presence of oscillations is important for pre-stimulus phase and amplitude regulation to occur, in models with a different type of critical phase transition (such as absorbing/sustained) (Del Papa et al., 2017; Levina et al., 2007).”

3) Dynamic Range.The dynamic range at criticality is larger than at sub/supercritical models. However, the "working point" or response range also changes considerably. This is in analogy to Zierenberg et al., 2020. How do your results relate on the systematic change of the response range, and to their ideas that combining networks of different distances to criticality would greatly enhance the dynamic range?

We thank the reviewers for bringing this study to our attention. We now address this point in the Discussion:

“We have shown that critical networks have overall the widest dynamic range of phase-locking response, and pre-stimulus amplitude and phase regulation. However, we also found that supercritical networks can show differential PLF and pre-stimulus regulation for the large stimulus sizes where the PLF response of critical networks saturates. One way in which the brain can take advantage of the information processing benefits of different states is to regulate its dynamics with respect to the critical point in a task-dependent manner (Pfeffer et al., 2018). Zierenberg et al., 2020, show that the brain could also combine networks with different distances to the critical point, such that the resulting ensemble has a dynamic range of stimulus response wider than any of the ensemble networks. This strategy could plausibly be employed to extend the range of stimuli strengths over which pre-stimulus amplitude and phase regulation occur.”

4) Writing.As to the writing, I think this paper is too concise and requires prior knowledge of the previously published papers from this group. I would appreciate more explanations in the main text about the key terminologies and concepts, such as "critical oscillation", "Long-range temporal correlations", "Detrended fluctuation analysis".

Following the reviewers’ suggestion, we have adjusted the Introduction to define these concepts.

“neuronal oscillations in the alpha band exhibit spontaneous amplitude fluctuations. However, these fluctuations are not random, but exhibit a specific temporal structure: autocorrelations with scale-free decay that extends up to time scales of hundreds of seconds (Linkenkaer-Hansen et al., 2001), also called long-range temporal correlations (LRTC). This scale-free character of the oscillations emerges in neuronal networks with balanced excitatory and inhibitory forces (Poil et al., 2012) and is considered a sign of critical-state dynamics (Bak et al., 1987; Chialvo, 2010). Thus, oscillations exhibiting LRTC are referred to as “critical oscillations”.”

“To evaluate the criticality of alpha oscillations, we used detrended fluctuation analysis (DFA, see Materials and methods), which can discriminate between random fluctuations in the amplitude of oscillations and critical oscillations with long-range temporal correlations.”

5) Data and code sharing.The raw spike data for the three states (sub/super/critical) should be made available, so that everyone can re-analyse them. The code should be made available as well. Please share the code before acceptance.

The code has been uploaded at:

https://figshare.com/articles/Pre-stimulus_phase_and_amplitude_regulation_of_phase-locked_responses_is_maximized_in_the_critical_state/12167943

6) I didn't find a clear definition of the stimulus input? What's the duration and magnitude? Is it a constant current in time? Why is the stimulus strength defined as the number of stimulated neurons? Would results be similar if stimulus strength is modeled as the magnitude of input current?

We thank the reviewers for suggesting to improve the definition of the stimulus input. We now clearly define the stimulus input in the Materials and methods:

“The stimulus had the same weight as any of the excitatory-excitatory connections in the network (w*_input_*= w*_EE_*= 0.0085), and was applied simultaneously (i.e., for 1 time step of 1 ms) to all the *n* excitatory neurons that the stimulus connected to. Additionally, we investigated the effect of the stimulus strength on network dynamics by modulating w*_input_* by a factor of 1/64 to 64 times the original value, on a log2 scale.”

We have also added this definition to the Results section:

“The stimulus had the same weight as any of the excitatory-to-excitatory synapses in the network (w*_input_*= w*_EE_*= 0.0085), and lasted for one time step (1 ms).”

7) Figure 1D, if input current is larger, does it expand the dynamic range of the supercritical networks?

To answer this question, we now explain why the PLF does not reach the maximum value of 1 in our network. In addition, we performed simulation experiments to test stronger stimulus strengths and report the results. We made the following changes to the manuscript (Results):

“Interestingly, in the case of supercritical networks, the strong, ongoing oscillatory activity prevents the phase-locking response from saturating (PLF~1), even at the largest stimulus sizes (Figure 1D, Figure 2—figure supplement 1).”

We acknowledge that another way to affect stimulus response is to alter stimulus strength (Results):

“Another factor that can shape stimulus response, other than stimulus size, is the strength of the external stimulus.”

Subsequently, we report the effect of changing the external input weight on the dynamic range (Results):

“We found that an 8-fold increase in the external input strength was sufficient to bring the PLF of supercritical networks to saturation, for the largest stimulus size (Figure 1—figure supplement 3B). However, such a large increase in the input strength also leads to a stronger PLF for the smaller stimulus sizes in the case of subcritical and critical networks, which means they no longer show weak PLF responses reducing their dynamic range. Of all tested strengths and networks, the maximal dynamic range occurs for critical networks at a multiplier ~1 (Figure 1—figure supplement 3C).”

8) Since previous work has shown criticality maximizes dynamic range of evoked rate responses, how is this result different from previous work?

The primary difference lies in the oscillatory nature of ongoing activity as well as of the stimulus-evoked activity in the CROS model. These properties are closely related to research using mass-neuronal electrophysiological readouts such as M/EEG. To emphasize these points, we have added new text and figures. In particular, (Results):

“Previously, it was shown that criticality maximizes the dynamic range of evoked firing-rate responses (Gautam et al., 2015; Kinouchi and Copelli, 2006; Larremore et al., 2011; Shew et al., 2009).”

Importantly, the CROS model produces EEG-like network oscillations and oscillatory responses to stimuli and, therefore, are suitably investigated for its phase-locking response as now explained (Results):

“However, event-related responses in large oscillatory networks also reflect a reset in the oscillatory phase following stimulation, leading to the phase-locking of the oscillation to the stimulus (Makeig et al., 2002; Min et al., 2007; Palva et al., 2005).”

To better visualize the oscillatory nature of the network, we have added raster plots of a single-trial stimulus response (as requested also in point #1 above) (Results):

“the stimulus response in our networks is indeed oscillatory, which may result in stimulus phase-locking (Figure 1B)”

Subsequently, we use the phase-locking factor to show how strongly different networks phase-lock to the stimulus (Results):

“Figure 2A: Ongoing oscillations phase-lock to stimulus. Example shown for 5 neurons stimulated, for sub-critical (*blue*), critical (*green*), and super-critical networks (*red*).”

Figure 1C, Is the phase-locking response related to the onset increase of firing rate? If the network shows onset response in rate, it would result in more rising phases, right?

As mentioned in our replies above, it is not ideal to use firing-rate when investigating oscillatory phenomena. To further clarify this, we included the following paragraph (Results):

“As such, the firing rate and phase-locking responses reflect fundamentally different aspects of network activity, e.g., evoked firing-rate responses may increase the overall network activity without affecting the phase of oscillatory activity in a particular frequency band. In fact, evoked firing-rate responses are not an appropriate measure of network response in networks with strong oscillations: they fluctuate continuously with the phase of the ongoing oscillation, making it difficult to judge the firing-rate responses in absolute values, i.e., without relating it to the ongoing oscillation. In that respect, the phase-locking response is a more appropriate way to detect changes in network activity occurring due to external stimulation”

Although we consider these valid arguments for why we analyze phase-locking responses instead of increases in firing rates, we have also included below an extensive analysis showing how firing-rate responses can be confounded with the oscillatory activity of the network, and confirming that firing rates are not always coupled to oscillatory responses in our model. We decided to not include these analysis results in the revised manuscript because the CROS model is oscillatory as emphasized in our replies above. We organized the following section as follows: first a short summary of our findings, followed by the corresponding figures, and end with a Materials and methods section. We begin by reiterating the reviewers’ question—"Figure 1C, Is the phase-locking response related to the onset increase of firing rate? If the network shows onset response in rate, it would result in more rising phases, right?”

– Yes, both increases and decreases in firing rate will cause significant phase-locking (Author response image 2), and they do so even in the absence of stimulation. Firing rates fluctuate with the ongoing oscillation, such that increases/decreases in firing rate correspond to specific oscillation phases, leading to high phase-locking factors.

**Author response image 2. respfig2:** Change in firing rate corresponds to the phase of the alpha oscillation for networks with strong alpha oscillations in the absence of stimulation. (**A**) Trial-averaged baseline-corrected firing rates corresponding to the largest decrease (blue), no substantial change (green), and highest increases (red) in firing rate. Intervals for calculating pre-stimulus baseline and post-stimulus firing rate are shaded in grey. (**B**) PLF at 150 ms post-stimulus (red shaded line) is high for both the lower and the higher firing-rate percentile bin, in the absence of any stimulation. Horizontal dashed line represents PLF significance threshold (see Materials and methods).

– However, the firing rate is not always coupled with the amplitude of alpha oscillations (Author response image 3). This is important, because both the ongoing activity, and also the stimulus responses, are oscillatory in nature. Thus, it is possible for a network to respond only with a change in the firing rate, without an oscillatory signal conducive to a change in the phase of the ongoing alpha oscillation (which is the main organizational principle in this network), and vice versa. Therefore, the phase-locking response we are measuring here is very different from the firing-rate response that has been measured in previous studies.

– Last, we investigated if the pre-stimulus firing rate regulates post-stimulus phase-locking (Author response image 4). As we have already shown in Author response image 2, the firing rate is strongly related to the oscillation phase. In our model, for networks with strong oscillations the phase of the oscillation diverges slowly, so the firing rate (oscillation phase) at a pre-stimulus time is predictive of the oscillation phase at post-stimulus time. As such pre-stimulus firing rate appears, inadvertently, to regulate post-stimulus *PLF* in the absence of stimuli Author response image 4). However, pre-stimulus amplitude does not have this issue: there is no regulation in unstimulated networks (Author response image 4).

**Author response image 3. respfig3:** Firing rates and amplitude of alpha oscillations are not always coupled. (**A**) For unstimulated networks, alpha-oscillation amplitude and firing rate are only strongly coupled in the critical regime. Pearson correlation of mean amplitude and firing rate calculated on 100 ms windows for unstimulated networks. (**B,C**) Coupling between post-stimulus oscillatory amplitude and firing rate is dependent on stimulus size. Pearson’s correlation coefficient between the firing rate (1–50 ms), and oscillation amplitude for the same time interval. For the weak stimulus (B, n = 5), the critical networks show stronger coupling between firing rate and amplitude compared to their sub/supercritical counterparts. However, for the strong stimulus (n = 1800) critical networks show weaker coupling between firing rate and amplitude, compared to their sub-/super-critical counterparts. (**D**) Coupling of post-stimulus amplitude and firing rate for the 3 example networks highlighted with triangles on the phase space (A-C), for the entire range of stimulus sizes.

**Author response image 4. respfig4:** Networks with strong oscillatory activity show regulation of phase-locking by pre-stimulus firing rate, even when the networks are not stimulated. (**A**) Trial-averaged amplitude for unstimulated networks, with trials split into lowest (blue), medium (green) and highest (red) pre-stimulus (grey window) oscillation amplitude. (**B**) No significant phase-locking for trials used in (**A**) in the unstimulated condition. (**C**) Trial-averaged firing rate for unstimulated networks, with trials split into lowest (blue), medium (green) and highest (red) pre-stimulus (grey window) firing rate. (**D**) Significant phase-locking for trials used in (**C**) in the unstimulated condition. Red dashed lines indicate post-stimulus time (equivalent to 25 ms post-stimulus corrected for filter effect), horizontal black dashed lines indicate PLF significance threshold (see Materials and methods).

Materials and methods:

PLF significance threshold

To determine whether the post-stimulus PLF for a particular pre-stimulus firing rate or pre-stimulus amplitude percentile bin is significant, we generated 200 random phases equal to the number of trials in a percentile bin. We repeated this process 10^6^ times. PLF values that are above the 95% percentile of the bootstrapped PLFs are considered significant.

Onset increase in firing rate and phase-locking factors

To investigate the relationship between the phase-locking response and the onset increase in firing rate, we chose a supercritical network which exhibits strong alpha oscillations. We ran the network without any stimulation, to investigate whether changes in firing rate can lead to phase-locking, also in unstimulated networks. As such, we analyzed the network as if it were stimulated, by generating a time series of “stimuli”, with the same temporal properties as the runs with stimulation (stimulus spacing at 1 second, with a 500-ms jitter). To measure the onset increase in firing rate, we quantified the firing rate at 1–50 ms post-“stimulus” relative to the firing rate during -275 to -175 ms pre-“stimulus” (corresponding to the -150 to -50 ms pre-stimulus interval used to study pre-stimulus amplitude regulation, after accounting for the 125 ms filter delay). We binned trials in 10 percentile bins based on the difference between post and pre-“stimulus” firing rate. Afterwards, we compared the average firing rate across trials in each bin, with the pre-stimulus firing rate activity subtracted. Signals were filtered in the 8–16 Hz frequency range, and PLF was computed across trials belonging to the same firing-rate percentile bin. Note that, due to the filter delay, the PLF time series (Author response image 2) is time shifted by 125 ms compared to the firing rate time series (Author response image 2).

Relationship between firing rate and amplitude of alpha oscillations

For unstimulated networks, we split the 2000 seconds time series into 100 ms windows. We then computed the mean firing rate and mean amplitude of alpha oscillations (8–16 Hz) for each of these short windows, and then correlated the two measures (Pearson). Amplitude windows were time shifted by 125 ms to account for the filter delay. For stimulated networks, we computed the average firing rate between 1 and 50 ms post-stimulus, and the average amplitude at 126 ms to 175 ms post-stimulus, again, accounting for the filter delay. We then correlated firing rate responses with post-stimulus amplitude using Pearson’s correlation for all networks / stimulus sizes.

Pre-stimulus firing-rate regulation of post-stimulus phase-locking response

We ran a supercritical network, without stimulation, for 2000 seconds. We generated a time series of “stimuli” (stimulus spacing at 1 second, 500 ms jitter), to analyze the network as if it were stimulated. To investigate the effect of pre-stimulus amplitude on post-stimulus phase locking, we split the trials in 10 percentile bins, based on the pre-stimulus amplitude between -150 and -50 ms (*grey rectangle*), and show the results for the 0–10^th^, 40–50^th^ and 90–100^th^ percentiles. We then computed the phase-locking factor across trials in each bin. To investigate the effect of pre-stimulus firing rate on post-stimulus phase locking, we again split the trials in 10 percentile bins, based on the pre-stimulus firing rate between -275 and -175 ms (*grey rectangle*), and show the results for the 0–10^th^, 40–50^th^ and 90–100^th^ percentiles. We then computed the phase-locking factor across trials in each bin. The 125 ms time difference between the pre-stimulus interval for firing rate and amplitude accounts for the filter delay. Also, note that, due to the filter delay, the PLF time series (Author response image 4) is time shifted by 125 ms compared to the firing rate time series (Author response image 4).

9) Figure 3D, Similar to my comments above about the dynamics range, I wonder if the range of pre-stimulus regulation depends on the magnitude of input current. If you drive the supercritical network with stronger input for each neuron, does it expand the region of significant regulation.

We now address this in the Results section, for pre-stimulus amplitude regulation:

“Changing the weight of the external input also affects the range of stimuli over which networks show pre-stimulus amplitude regulation (Figure 2—figure supplement 3B).Nonetheless, the range of stimuli over which the critical networks show pre-stimulus amplitude regulation is superior to that of the subcritical and supercritical networks, regardless of the strength of the input weight (Figure 2—figure supplement 3C).”

And also for pre-stimulus phase regulation:

“We found that changing the weight of the external input also affects the range of stimuli over which pre-stimulus phase regulation is significant (Figure 3—figure supplement 3B). Nonetheless, pre-stimulus phase-regulation occurs for a wider range of stimuli in critical networks than in their subcritical or supercritical counterparts, regardless of the strength of the input weight (Figure 3—figure supplement 3C).”

10) About the balance between excitation and inhibition: I think it should be the product of synaptic weight and connection probability that determines the network dynamics. Why is the critical transition at approximately when excitatory connectivity percentage equals inhibitory connectivity percentage? How does the transition depends on the recurrent weights?

As the reviewers correctly point out, the balance of excitation and inhibition is indeed a bi-product of connectivity (number of connections) and the strength of the recurrent weights. The critical line follows a diagonal through the phase space where excitatory and inhibitory connectivity are roughly equal, but the exact positioning of the line is coincidental, i.e., the optimization algorithm could in principle have selected the parameters shown in Figure 4 of the revised manuscript. We showed, when addressing Major Point 2, that by modulating the weight of excitatory and excitatory connections, we can shift the position of the critical line on the phase space, and its position does not affect the relationship between criticality and versatility of information processing (Results):

“Therefore, we expect that the functional consequences of criticality on stimulus processing that emerged out of the model, are generic to criticality, and not specific to the actual parameters. To verify this, we repeated our analysis for two versions of the model where the synaptic weight of excitatory to excitatory connections (w*_EE_*) was multiplied by a factor of 0.75, or by a factor of 1.25. Multiplication of w*_EE_* by a factor of 0.75 reduced the overall activity in the network, which means that stronger connectivity is required to achieve the same dynamics, leading to a shift up of the critical line through the phase space (Figure 4). Increasing w*_EE_* had the opposite effect on the critical line. Importantly, the functional consequences of criticality—maximized dynamic range, pre-stimulus amplitude and phase regulation—followed this shift in the critical line. This suggests that the relationship between criticality and versatility in stimulus processing is not coincidental on the specific choice of parameters, but generic to criticality.”